# Design of a Reverse Logistics System with Internet of Things for Service Parts Management

**Daniel Y. Mo** [1],*[ID]**, Chris Y. T. Ma** [2][ID]**, Danny C. K. Ho** [1][ID] **and Yue Wang** [1]

1   Department of Supply Chain and Information Management, The Hang Seng University of Hong Kong, Hong Kong, China
2   Department of Computing, The Hang Seng University of Hong Kong, Hong Kong, China
*   Correspondence: danielmo@hsu.edu.hk

**Abstract:** Despite that reverse logistics of service parts enables the reuse of failed components to achieve greater environmental and economic benefits, the research and successful business cases are inadequate. This study designs a novel reverse logistics system that applies the Internet of Things (IoT) and business intelligence to streamline the reverse logistics process by identifying the appropriate components for sustainable operations of component reuse. Furthermore, an inventory classification scheme and an analytical model are developed to identify the failed components for refurbishment by considering return quantity of the failed component, repair rate of the failed component in the repairing center, reusable rate of refurbished parts, corresponding costs, and the benefit of refurbished parts. Moreover, a mobile application powered by the IoT technology is developed to streamline the process flow and avoid collection of fake components. Lastly, a case study of an electronic product company is conducted, and it is concluded that the proposed approach enabled the company to facilitate the reuse of components and achieve the benefit of cost saving. The results of this study demonstrate the importance of a reverse logistics system for companies to sustain after-market service operations.

**Keywords:** reverse logistics; service parts management; analytical model

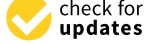



## 1. Introduction

Service parts management for critical systems, such as high-tech systems, medical equipment, and office equipment, has developed worldwide [1]. The management of service parts is crucial to enhance service quality, sustain products, and capture business opportunities across various sectors. Managing serviceable items becomes a common industrial practice nowadays. However, it is challenging to ensure that the required parts are made available at appropriate points in the supply chain to achieve the desired service level [2], owing to hurdles, such as a large variety of parts, risk of stock obsolescence [3], irregular or lumpy demand patterns characterizing many parts [4], and high responsiveness [5]. An added complexity occurs when there is a strong need for effective refurbishing failed components to improve corporate environmental performance. Although the adoption of advanced information and communication technologies and a holistic approach for connecting both the forward and reverse logistics of components are expected to sustain operations that deliver greater economic and environmental benefits, more research studies and successful business cases are required.

To address the challenges and research gap in sustainable service parts management, this study aims to propose a design framework of an integrated reverse logistics system powered by the Internet of Things (IoT) and business intelligence (BI) to enable fast and smooth forward and reverse flows for refurbishing failed components. The proposed system is unique on several fronts. Firstly, it addresses the characteristics of high variety and demand uncertainty of service parts, which is common in practice, by analyzing parts usage



to identify the appropriate reusable parts and avoid unnecessary refurbishment. Secondly, to sustain the refurbishing operations of failed components from an economic perspective, a BI-enabled analytical model is developed to evaluate the operation requirements of refurbishing failed components by considering factors of return quantity of failed components, repair rate of failed components in the repairing center, reusable rate of refurbished parts, corresponding costs, and the benefit of refurbished parts. Thirdly, a mobile application solution powered by IoT is proposed to streamline the reverse logistics process and avoid collecting fake components. Lastly, the functionalities of the proposed reverse logistics system are verified by conducting a case study on a leading electronic product company in Hong Kong. The analysis of data collected from the collaborative company's enterprise resource planning system showed over 200 stocking units could be reused to avoid material waste and enhance the process flow of return parts for service parts management. Hence, the proposed design framework contributes to not only filling the research gap in literature but also providing an industrial guideline for service parts operations.

The remainder of this paper is organized as follows. Section 2 presents a literature review on service parts management, BI with IoT, and reverse logistics systems. Section 3 describes the design framework of an integrated reverse logistics system with IoT for service parts management. Furthermore, an analytical model is presented to identify the break-even relationship between the benefit of component reuse and rates of return, reverse logistics, and repair to sustain reverse logistics and component reuse operations. Section 4 presents a case study to demonstrate the benefits of the proposed reverse logistics system with IoT. Finally, Section 5 concludes the study and identifies areas for future research.

## 2. Literature Review

To better design a reverse logistics system powered by IoT and BI for service parts management, this study has reviewed and integrated past relevant research in the domains of service parts management, BI with IoT, and reverse logistics systems.

### 2.1. Service Parts Management

Management of service parts is a subset of system maintenance and can be categorized into two types: break-fix maintenance and preventative maintenance. Analysis of the overall maintenance costs between break-fix and preventative maintenance [6] shows that the maintenance costs of a repair performed in the reactive mode of the break-fix is approximately three times higher than in preventive mode. Although preventative maintenance is an advanced method of ensuring the availability of critical systems, predicting the error of machine failures remains a major challenge. Moreover, information integration with the downstream supply chain is almost absent. Cassivi [7] emphasized the importance of widespread collaboration in forecasting and replenishing supply chain partners. In supply chain collaboration, Boone et al. [8] commented that most companies lacked a system perspective for service parts management, owing to weak supplier relationship management and inaccurate demand forecasts. Through interviews with several service parts managers, Boone et al. [9] identified the downstream requirements of service parts planning, forecasting, and outbound distribution. It was concluded that, although service parts management can be enhanced through preventive maintenance with improved prediction, information integration with supply chain partners was the core element. Furthermore, distribution of service parts needed to respond to customer needs. For example, repair services for medical equipment often require quick responses within hours rather than days.

Garcia et al. [10] proposed a system that continuously monitored equipment performance and analyzed deviations of sub-cycle times, which provided a signal of potential failures to support the scheduling of preventive maintenance before machines failed. Therefore, the industry needs to further integrate its downstream supply chain partners and perform better preventive maintenance using advanced technologies, such as IoT, machine learning, and big data in Industry 4.0. [11]. Moreover, IoT can be used not only to enhance operational efficiency but also to improve the reuse of components for sustainability, consid-

ering that increased operational efficiency would result in substantial cost savings [12,13] for supporting the system integration of downstream supply chain partners. Thanks to the system integration of downstream supply chain partners for information sharing [9], smarter decisions of handling ad hoc breakdown issues by inventories in real time [14] and obtaining failed components for reuse and recycling can be achieved. Having a service parts management system with a network of IoT-enabled devices that contain sensors for data collection and exchange per se is inadequate. To realize business value from big data, IoT solutions need to be connected to BI. However, studies on the application of BI with IoT in reverse logistics systems for service parts management are insufficient.

### 2.2. Business Intelligence with Internet of Things

BI systems are data-driven, enterprise-wide decision-support systems that integrate data gathering and storage with advanced analytical functions for decision making [15]. They are used to enhance managerial decisions, visualize operations status, recommend operational decisions, and facilitate collaboration with supply chain partners through IoT and incorporate the processes and systems that visualize operation performance systematically [16]. For example, online analytical processing (OLAP) tools are used to report, analyze, model, and plan for business optimization [17]. Elements of a BI system include operational source systems, data-staging modules, data presentation modules, and data access tools [18]. An operational source system is used to store and analyze historical data. The data-staging module comprises both the storage area and a set of processes called extract-transform-load. The data presentation module is a data warehouse where data are organized and accessed by other systems. Data access tools use data stored in the presentation area to visualize information [18]. Furthermore, to support managerial decisions, information from data warehouses and relational databases is presented to managers in the form of dashboards, scorecards, and reports [19]. The dashboard not only aligns the operations of the company with their business goals but also visualizes key operational status to improve decision-making processes to enhance competitive advantages [20]. However, traditional BI systems lack system integration with supply chain partners, and do not proactively respond to situations and support critical business decisions in real time to optimize business processes [17].

Given that customers' expectations of service lead time and service level are higher than ever, real-time supply chain visibility via mobile solutions is necessary to ensure efficient operations, especially for service parts management [5]. IoT and mobile technologies are fundamental information and communication technologies [21] that support the BI process for managers to make strategic decisions and achieve effective planning across space in real time [22]. This information sharing among decision makers can be achieved with IoT and mobile technologies [14]. Electronic devices, such as barcode systems and radio frequency identification (RFID) systems, are commonly used to connect physical objects with virtual representations. In the logistics industry, RFID-supported devices are used to identify individual items using RFID tags and communicate with logistics systems for tracking and tracing [23]. Akbari et al. [24] stated that supply chain operational efficiency can be enhanced using RFID technology to facilitate communication between data in an item and information in the distributed database. Compared to barcodes, RFID technology has become more popular for service parts management because it can identify an item in a low-light environment without direct contact [24]. To connect product information with supply chain information systems for BI, mobile technology is another crucial element, which comprises portable two-way communication devices, computing devices, and networking technology. Managers can obtain real-time information via mobile technology to make real-time decisions [25]. The recent information technologies in Industry 4.0 are focused on the integration of IoT, machine learning, blockchain, etc. [26–29]. For the reverse logistics systems proposed in this study, managers can promptly trace products for return and reuse.

### 2.3. Reverse Logistics Systems

By integrating reverse logistics into the conventional supply chain process, the development of closed-loop supply chains has been receiving wide attention across different contexts, such as recycling and remanufacturing, to achieve sustainability [30]. Product design incorporating component reuse is key for remanufacturing [31], whereas analyzing the cost structure serves as the fundamental analytical model to enable component reuse in a product family for remanufacturing [32]. In the pursuit of sustainable operations of component reuse, service parts inventory management presents high potential and opportunities for expanding our focus, from improving the forward flow of the required parts for repair and maintenance [33] to developing reverse logistics systems for higher utilization of service parts, while reducing waste and cost. This is highly relevant in practice, as overstocking occurs due to the difficulty of accurately predicting the required number of service parts with nonstationary demand. In addition, these parts will eventually become dead stock and hence, waste if left unmanaged. To address this fundamental problem in a closed-loop logistics network, the redeployment strategy of service parts through reverse logistics in large-scale and multi-echelon operations has been investigated and considered an effective method [34].

Although a close relationship exists between reverse logistics and service parts inventory management, research in this area is inadequate [35]. A promising way to fill this research gap is to examine the application of Industry 4.0 technologies in sustainable logistics [36] to ensure better spare parts inventory management. One approach for waste reduction is to apply various additive manufacturing technologies to produce low-volume and highly customized spare parts by adding layers of materials together when needed. However, issues such as intellectual property and liability must be addressed to ensure the approach is widely accepted [37]. Another approach for the pursuit of sustainable spare parts inventory management is to apply the IoT to inventory management [38]. To date, this study builds on previous works on the redeployment of excess inventory [34] and BI [39] and develops an IoT-enabled reverse logistics system with the connection of RFID tags to a cloud computing system to enable a mobile decision support system for timely identification and effective deployment of reusable parts. The proposed design of a reverse logistics system focuses on the interactions between users and an information system in a business process under a service-oriented architecture, which is adopted to the typical design of a logistics system [40]. The proposed IoT solution was tested in a company and achieved benefits of higher spare parts utilization, cost savings, and reduction of material waste. The following sections present the details of the system.

## 3. Design of a Reverse Logistics System with Internet of Things for Service Parts Management

The proposed system is designed to achieve three functionalities. Firstly, the system needs to achieve fast and smooth item flows in both forward and reverse directions for service parts replacement and return. Secondly, the system needs to identify reusable items for reverse logistics. Thirdly, the system needs to enable information sharing among decision makers in real time.

While designing the proposed reverse logistics system with the IoT for service parts management, it is essential to identify the characteristics of service parts management from the supply-chain-planning perspective. Compared to the forward logistics planning of products, the characteristics of service parts planning include a high variety of stocking units, low usage of parts, high criticality of stock-out, and long component life cycle, making it difficult to identify and determine the appropriate volume of components for reuse and/or recycling. To address this problem, the current study proposes an item classification analysis framework using BI methods to predict the return rates of components more accurately. Although item classification is useful for both inventory planning and prediction of returnable components, the volume of returnable and reusable components can vary, owing to factors such as component life cycle, disassembly of components, and

overall reverse logistics costs. Therefore, an analytical model is critical for determining the set of reusable components for sustainability purposes. Furthermore, a mobile application with IoT is proposed to streamline the entire reverse logistics process, such that counterfeit components are avoided to minimize reverse logistics costs and safeguard improper replacement outside the agreed service scope. The following sections describe the details of the three major elements in the design of a reverse logistics system with IoT.

### 3.1. Process Flow of Service Parts Replacement and Return

The value of the service parts is to sustain the critical systems and operations performance by replacing the failed components in a timely manner. In a narrow view, service parts management focuses on the replacement of service parts for system maintenance. In a broader view, the performance of service parts management affects the sustainability of critical systems and the corresponding components, which includes the reverse logistics management of failed components returned for repair. The proposed service parts management includes the management of items and information in both forward and reverse logistics flows.

Figure 1 shows the process flow of service parts replacement and return. Based on the service-oriented architecture [40], it is essential to identify user interactions with the system, which includes network service and applications, in the business process. In the proposed system, service parts are stored in a warehouse and used to fulfill part-replacement order requests. Parts and product information, such as historical usage, bill of materials, and serial number, are stored in the cloud database with network service and are accessed by users via a mobile application. When a parts replacement order is requested, the corresponding service part in the warehouse is delivered to the on-site equipment location for replacement. Considering the importance of identifying whether the failed component is provided by the original manufacturer, the RFID tag attached to the failed component must be authenticated using one of the various RFID tag authentication approaches [41–43]. Here, we use "encrypted product ID" to indicate that such a scheme is deployed to authenticate the tag and prevent tag cloning (i.e., a counterfeit component). Encrypted production ID information is stored in an RFID tag for each item. The failed component is replaced after item verification and returned to the warehouse for further processing. The returned parts in the warehouse are then disposed or delivered to the repair center for reuse and/or recycling. The reusability of returned parts is determined by the predicted usage of service parts, conditions of failed components, availability of service parts in the market, and total logistics and repairing costs. Furthermore, the delivery decision of returned parts to the repair center is made by considering the predicted usage of service parts. If the returned component is obsolete or becomes excessive in the operations, it is directly disposed in the warehouse to avoid unnecessary waste of transportation and repair. Therefore, item classification of part usage is important for determining the reusability of service parts.

### 3.2. Item Classification by Business Intelligence

In this section, we propose a method of item classification for predicting the reusability rate of service parts. Our approach is similar to the classical ABC classification system where the top 80% of item usage is classified as class A items, 15% of item usage is classified as class B items, and the bottom 5% of item usage is classified as class C items. The grouping of the massive number of items by a limited number of classes aids the decision to return the appropriate failed component to the repair center.

Yet, when compared to the static item classification approach in classical supply-chain-planning systems, the proposed system dynamically updates the class of items using BI. The class type of the failed component is updated by the decision rules depending on the latest part usage information stored in the cloud database.

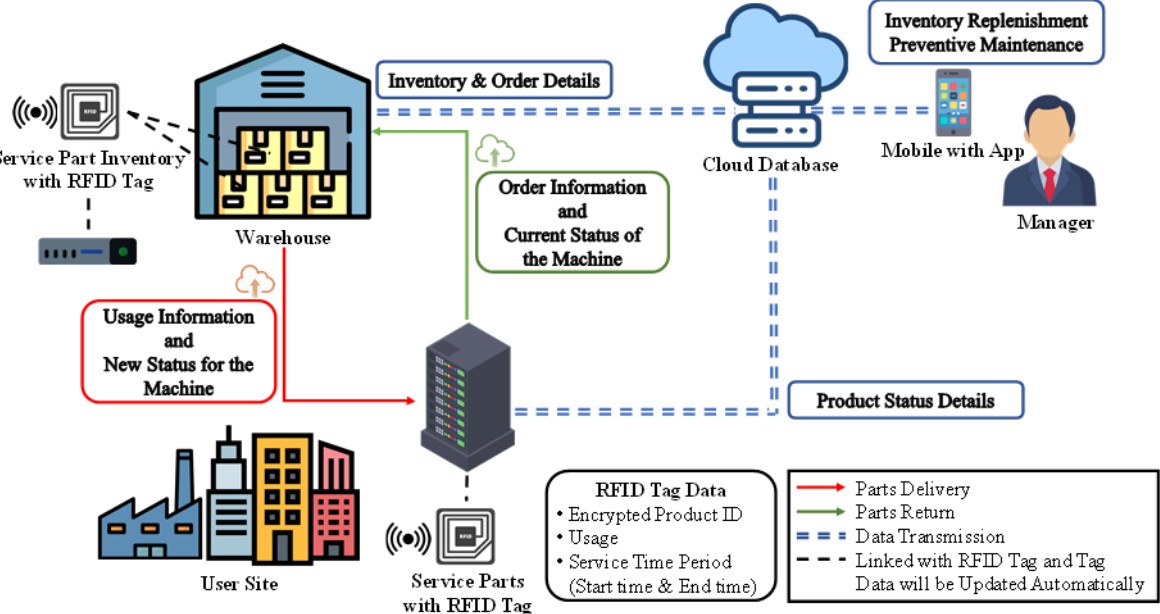

**Figure 1.** Service parts process flow for part replacement and return.

Figure 2 presents an example of item classification on a BI dashboard, which is accessed by the service-parts-planning manager. Class A items have the highest usage (45,788 units) of the three classes, representing over 80% of the total usage. However, the number of class A items is just 277, representing only 5.5% of the total items. The BI-enabled classification system is connected to an analytical model that considers the average usage rate of each class item to identify reusable items for reserve logistics.

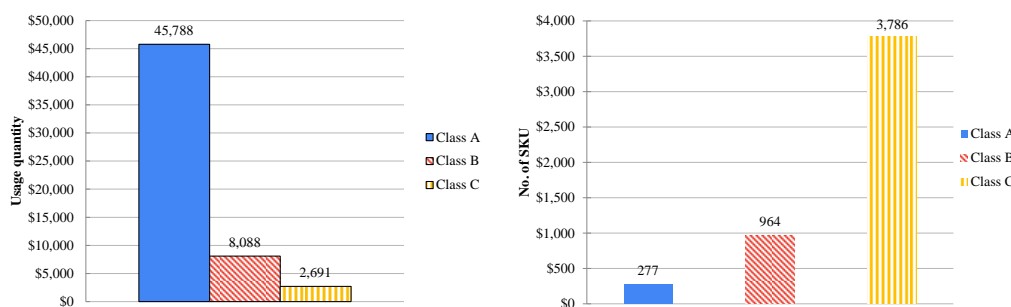

**Figure 2.** Usage and item units of each class.

### 3.3. Identification of Reusable Items for Reverse Logistics

Item classification of service parts facilitates the decision to determine failed components for reuse and repair. Reusability of failed components depends on several factors, including return quantity of failed components ($N$), return rate to the repair center ($\alpha$), repair rate in the repairing center ($\beta$), and reusable rate of the refurbished part ($\theta$).

Figure 3 shows the transformation process of the failed components into refurbished parts for reuse. First, different failed components are identified through a reverse logistics system. For each type of failed component, the quantity received at a given time is represented by $N$. After the initial screening, a portion of each type of failed component is shipped to the repair center for further assessment of repair. The portion of failed components flowing to the repair center is measured by return rate ($\alpha$) with the reverse logistics unit cost, $c_l$. Total reverse logistics cost is $N\alpha c_l$. The portion of non-repairable parts is measured by the rate of (1-$\alpha$), and these parts are disposed of with a unit disposal cost of $c_d$. For simplicity, $c_d$ is assumed to be the same at different stages. In the repair

center, the failed components are fully assessed for repairability. However, only a portion is transformed into refurbished parts for use. The repair rate of a particular type of failed component is denoted by $\beta$, and the corresponding quantity of refurbished parts is $\alpha\beta N$. Hence, the total refurbishing cost is denoted as $X(\alpha, \beta) = N(\alpha c_l + \alpha\beta c_r)$. To identify a component as a refurbished part, repair unit cost of $c_r$ is incurred. The non-repairable part is disposed at the cost of $c_d$. At the end of the transformation process, the refurbished part is made available for reuse.

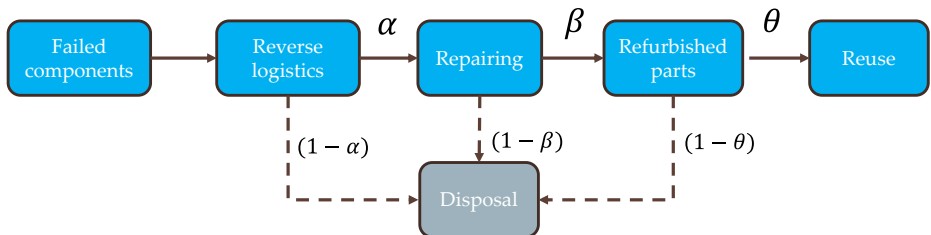

**Figure 3.** Return flow of the failed component for reuse.

The value of a refurbished part depends on whether it is used or becomes a waste of dead stock. For financial sustainability, the total cost of using a refurbished part should not exceed revenue generated from its usage. For the parts which cannot be refurbished or used, a disposal cost would incur. The total disposal cost is denoted as $Y(\alpha, \beta, \theta) = c_d[(1-\alpha)N + \alpha(1-\beta)N + \alpha\beta(1-\theta)N]$. The benefit of using a unit of the refurbished part is denoted by p, and the portion of a particular type of refurbished part being used is measured by the usage rate of $\theta$. The profit from using the refurbished part is the difference between the revenue $p\alpha\beta\theta N$ and the total costs of transformation (i.e., the sum of reverse logistics and repairing costs) and disposal. Hence, the overall profit is denoted as $Z(\alpha, \beta, \theta) = pN\alpha\beta\theta - X(\alpha, \beta) - Y(\alpha, \beta, \theta)$.

To sustain reverse logistics and repair operations for the reuse of service parts, determining the lowest acceptable level of $p$ is essential, given a set of return, repairing, and refurbishing rates of $\alpha, \beta$, and $\theta$ and transformation and disposal costs. The value of the benefit can be determined by break-even analysis:

$$
\begin{aligned}
Z(\alpha, \beta, \theta) \quad &= pN\alpha\beta\theta - X(\alpha, \beta) - Y(\alpha, \beta, \theta) > 0 \\
&\Rightarrow p > \frac{\alpha c_l + \alpha\beta c_r + c_d[(1-\alpha) + \alpha(1-\beta) + \alpha\beta(1-\theta)]}{\alpha\beta\theta}
\end{aligned}
\tag{1}
$$

Accordingly, Equation (1) provides the critical ratio of $\frac{\alpha c_l + \alpha\beta c_r + c_d[(1-\alpha) + \alpha(1-\beta) + \alpha\beta(1-\theta)]}{\alpha\beta\theta}$ to identify the required benefit of $p$ for sustainable reverse logistics of service parts. Moreover, to analyze how profit changes with return, repairing, and reusable rates, we need to take the first derivative of $Z$ with respect to $\alpha, \beta$, and $\theta$:

$$
\begin{aligned}
\frac{\partial Z(\alpha, \beta, \theta)}{\partial \alpha} &= pN\beta\theta - Nc_l - N\beta c_r + c_d N - c_d(1-\beta)N - c_d\beta(1-\theta)N \\
&= N(\beta\theta(p + c_d) - \beta c_r - c_l)
\end{aligned}
\tag{2}
$$

$$
\begin{aligned}
\frac{\partial Z(\alpha, \beta, \theta)}{\partial \beta} &= pN\alpha\theta - \alpha Nc_d + \alpha(1-\theta)Nc_d \\
&= N\alpha\theta(p - c_d)
\end{aligned}
\tag{3}
$$

$$
\begin{aligned}
\frac{\partial Z(\alpha, \beta, \theta)}{\partial \theta} &= pN\alpha\beta + \alpha\beta c_d N \\
&= N\alpha\beta(p + c_d)
\end{aligned}
\tag{4}
$$

The first derivatives of $Z(\alpha, \beta, \theta)$ refer to the conditions for increasing profits when the function values of Equations (2)–(4) are positive. Given that the values of $N, \alpha, \beta, \theta, p, c_d$, and $c_l$

are positive for any failed component to be refurbished, profit increases with $N$, $\alpha$, $\beta$, and $\theta$ when the following equations are satisfied:

$$c_l < \beta\theta(p + c_d) - \beta c_r \tag{5}$$

$$c_d < p \tag{6}$$

Equations (5) and (6) indicate the upper bound of the reverse logistics and disposal costs, respectively, for obtaining increasing profit with the corresponding return, repairing, and reusable rates. The relationship between the benefit of refurbished parts, return quantity of failed component, repair rate of failed component in the repairing center, reusable rate of refurbished part, and the corresponding costs are illustrated through a case study.

*3.4. Information Sharing through Mobile Application with Internet of Things*

The success of reverse logistics in the proposed framework depends on the availability and sharing of the required data in real time for making informed decisions to obtain optimized outcomes. To this end, the current study demonstrates that IoT-powered mobile applications are highly relevant and effective. For example, physical components with attached RFID tags are connected to their cyber (virtual) counterparts when the tags are read by a network-connected RFID reader. Such connections authenticate these physical components and their inventory management and information collection for BI systems to make decisions, such as the reusability of different failed components. Hence, a mobile application that can communicate and share information with the replaced parts and the cloud database is critical to the success of our reverse logistics framework.

To design the mobile application as a prototype, the unified modelling language (UML) of a use case diagram is used to demonstrate the steps of the application (Figure 4), which illustrates how the key mobile application functions would streamline the operational processes of forward and reverse logistics:

1. Forward logistics: Lists the components required in a part replacement order and ensures replacement.
2. Reverse logistics: Authenticates the replaced parts and checks their reusability with the cloud database.

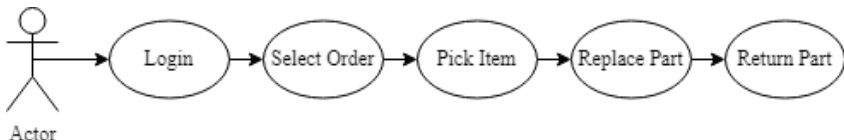

**Figure 4.** Use case diagram.

These functions are enabled by the mobile application as an RFID reader to communicate with the RFID tags attached to the components and exchange the collected information with a cloud database, which is used to share real-time information with various BI systems to facilitate reverse logistics decision-making.

## 4. Illustrative Case Study

The case study of a leading company of electronic appliances and products in Hong Kong was conducted to illustrate and evaluate the proposed approaches to reverse logistics for service parts management. The collaborating company provided a wide range of electronic products with warranty service options. The inventory-planning process for selling products and providing service parts replacement was managed by an enterprise-resource-planning (ERP) system. However, the reverse logistics process was decoupled from the ERP system, resulting in the wastage of excess service parts and unsustainable repair operations in the reuse of components. Furthermore, to investigate the operational

issue and explore the opportunity of reusing components, a dataset, including inventory, usage, and in-transit shipment reports of six months of operational data, was collected. Under the proposed design framework of reverse logistics systems, these reports were stored in a cloud database, and the reusability of the components was evaluated using the BI-powered analytical model, which in turn was used to visualize the inventory status. Subsequently, information sharing through a mobile application with IoT to streamline the reverse logistics flow was illustrated.

### 4.1. Evaluation of Reusable Items for Reverse Logistics

A BI dashboard was first developed to evaluate the reusability of components by item classification based on the collected operational reports. According to the raw data of inventory and usage reports, a BI indicator was used to identify the weeks of supply for each item class.

Figure 5 shows the part usage, inventory status, and weeks of supply per class under the proposed inventory classification. It was revealed that all class C items had excess inventories, given that the corresponding weeks-of-supply indicator showed that the inventory of those items would be used for over a year. Regarding class B items, most items had inventories for eight weeks of supply on average. After discussion with the general manager, these items were not initially selected for reuse. Instead, all class A items were regarded as appropriate components for reverse logistics and repair, considering these items were guaranteed for reuse. To apply the proposed analytical model, the cost and price parameters of $p$, $c_r$, $c_l$, and $c_d$ were obtained from the collaborating company. From the perspective of sustainable repair operations, evaluating how the return rate ($\alpha$) and repair rate ($\beta$) affect different costs and the overall profits is critical, based on Equation (1). For a numerical illustration, a set of operations and cost parameters is obtained as follows (Table 1):

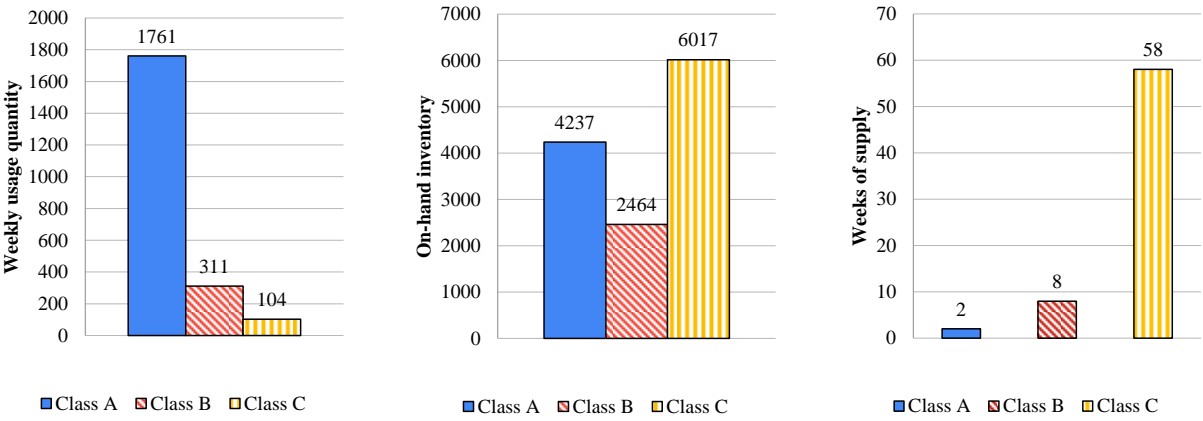

**Figure 5.** Visualization of parts usage and inventory status.

**Table 1.** Parameter for refurbishing operations.

| Parameter | Value |
| --- | --- |
| Return quantity of failed component ($N$) | 10 units |
| Return rate ($\alpha$) | [10%, 20%, . . . , 90%] |
| Repair rate ($\beta$) | [10%, 20%, . . . , 90%] |
| Usage rate ($\theta$) | 90% |
| Reverse logistics cost ($c_l$) | $10 |
| Repairing cost ($c_r$) | $100 |
| Disposal cost ($c_d$) | $50 |
| Cost saving of refurbished parts ($p$) | $200 |

Figure 6 shows how different types of costs and profit change when the return rate ($\alpha$) and repair rate ($\beta$) increase in the same rate. The repairing operation is sustainable with the positive profit only when $\alpha$ and $\beta$ are higher than 70%, considering Equation (1) has the quasi-convex property. When $\alpha$ and $\beta$ are equal to or higher than 30%, profit increases as return and repair rates increase. Furthermore, Equations (5) and (6) are both satisfied in that particular range, thereby demonstrating the analytical model's effectiveness in setting the targets of return rate and repair rate for achieving more sustainable operations. Moreover, sensitivity analysis would be conducted for cost management of refurbishing operations.

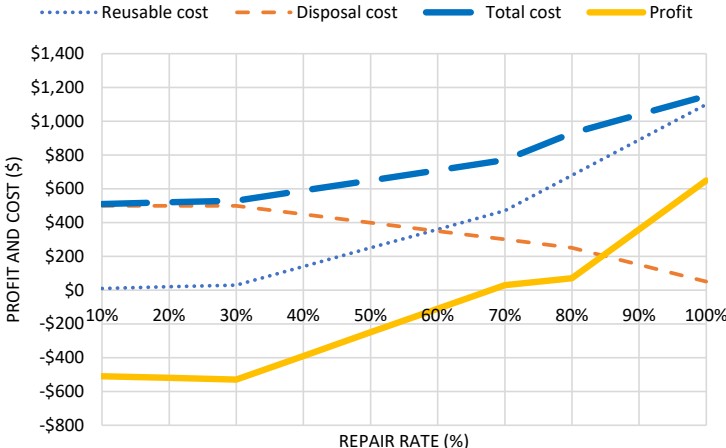

**Figure 6.** Visualization of part usage and inventory status.

Figure 7 shows the effects of disposal and repairing costs on profits. Considering the same range of disposal and repairing costs, an increase in repairing cost has a greater impact on the total cost and profit than an increase in disposal cost. Essentially, the profit is more sensitive to repairing costs but less sensitive to disposal costs, thereby indicating that the effectiveness of the repair operation is a requirement for sustainable operations.

To improve the repair operation, a mobile application solution with IoT is developed to streamline the reverse logistics process and avoid fake components for receiving the appropriate items and obtaining a higher repair rate.

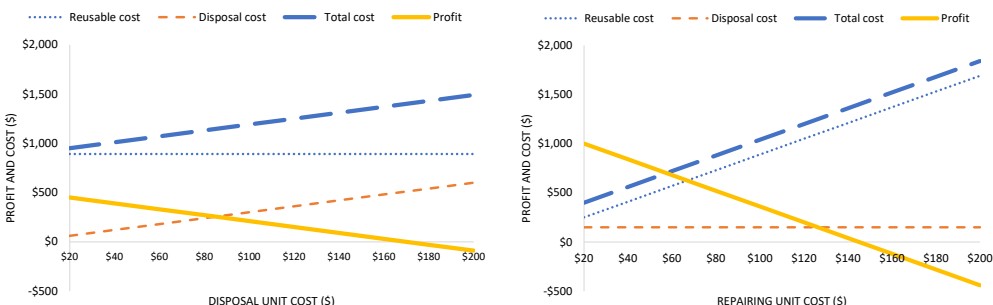

**Figure 7.** Sensitivity analysis of disposal cost and repairing cost.

### 4.2. Mobile Application Solution with Internet of Things

To streamline workflow for part replacement and return, the mobile application solution facilitates both forward and reverse logistics. Particularly, the operator uses the mobile application to select the replacement order to be fulfilled, collect the corresponding service parts, authenticate the replaced service parts, and decide whether the replaced service parts are to be refurbished. The details of the mobile application solution are presented below (Figure 8).

To complete the order, the operator must follow the steps shown in the mobile application.

Step 1. Operator logs in to the mobile application and accesses the parts-replacement order list.

Step 2. Operator selects an order from the list that provides the order details at the bottom part of the screen, such as the number of items for each part in the order.

Step 3. Operator follows the order and collects the required service parts from the warehouse using an RFID tag.

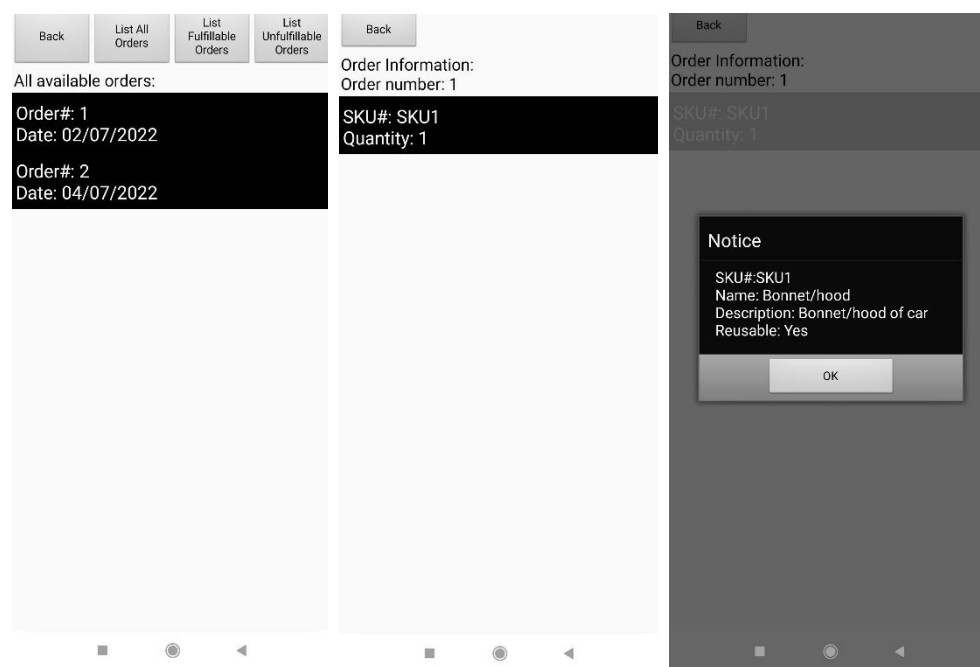

**Figure 8.** Replacement orders for returned items in mobile application.

Step 4. When the operator arrives at the site and begins replacement work, s/he uses the app to complete the order by scanning the RFID tag of the replacing item. When completing a parts replacement order, the corresponding replaced item must be matched with the same type of replacing item. When the replaced item is scanned, the app also checks its authenticity with the cloud database with an appropriate authentication scheme [39–41] and guides the operator on whether the replaced item should be returned for refurbishment.

Step 5. Once all items of the replacement order have been replaced and scanned successfully, the order is considered completed. The app then prompts the operator of its completion, together with the corresponding updates in the database.

As discussed in a related work [44], inexpensive rewritable tags without any computational abilities cannot prevent tag cloning. The proposed framework assumes the ability of RFID tags to generate random numbers and perform (keyed) hash function or cyclic redundancy code (CRC) following RFID tag authentication schemes [41–43] for authentication purposes. Otherwise, RFID tags without such abilities can potentially be cloned such that genuine parts can be substituted by counterfeit ones. However, the chance of cloning an RFID tag is low in the proposed reverse logistics system, owing to the lack of incentive for users to clone the RFID tag and hack the system.

## 5. Discussion

The proposed reverse logistics system integrates BI and IoT to streamline the process flow of collecting the failed components and reusing the refurbished components. The information of failed components gathered via IoT enables the decision of identifying the appropriate components for refurbishment and evaluating the expected quantity of component reuse by the analytical model. This integrated approach transforms the typical forward logistics flow of service parts to a closed-loop service parts logistics flow for attaining a sustainable operation. The benefits of component reuse and cost-saving opportunities are illustrated via a case study. Yet, the proposed design has some limitations. To achieve its full potential, some major hurdles in the implementation process need to be addressed and resolved in practice. Firstly, an integration between the inventory-planning system and the reverse logistics system is required to enable a fully automated decision-making process. If data collection and sharing between systems are not done seamlessly, an extensive lag time is expected. The latency could lead to suboptimized decisions due to the potential discrepancy between the primary and replicate data. Secondly, the information stored in the RFID tag can be cloned by hacking technology and this would result in the decrease of the refurbishing rate of the component reuse. The above-mentioned-RFID tag authentication schemes are recommended for the system that manages high-value service parts. Yet, these advanced solutions may demand more technical support and increase implementation cost and time. Thirdly, the current analytical model is based on the expected return quantity of failed components without considering the uncertainty of component return over time. This may cause a delay in the refurbishment process if the return of failed component fluctuates dramatically. A stochastic analytical model would be needed for better planning in these situations. Additionally, that should be addressed in the future development of more sophisticated models that can be applied in different contexts.

## 6. Conclusions

Service parts management has become more important for companies seeking economic benefits and reusing failed components through reverse logistics for sustainability. This study proposed and examined a holistic design of reverse logistics systems using analytical models powered by BI and IoT. The proposed analytical models built the much-needed link between forward and reverse logistics to identify appropriate failed components for reuse. The relationship among reverse logistics quantity, return rate, repair rate, reusable rate of refurbished part, and corresponding costs were formulated into the analytical model to provide managerial guidelines for sustaining reverse logistics and repair operations. To streamline the operations, a mobile application with IoT was proposed. The mobile application, which accessed component information through an RFID tag and shared real-time information through cloud technology, allowed managers to avoid the collection of fake components and ensure sustainable operations. The proposed reverse logistics system was illustrated through a case study of a leading company of electronic appliances and products in Hong Kong. Over 200 stocking units were identified for the reuse of components. The guidelines of expected return rate, disposal cost, and repairing

cost controls developed by the analytical model for the company were effective to sustain operations for higher economic and environmental gains.

These results highlight the design of a reverse logistics system for future studies along two directions to address the limitations as discussed above. First, enhancing the analytical model by considering the demand uncertainty for both forward logistics inventory and reverse logistics operation management is essential. Compared to the deterministic analytical model without considering demand uncertainty, stochastic analytical models provide opportunities for higher cost savings and larger quantities of component reuses. Second, to reduce the risk of fake components, the security of RFID tags can be further advanced using different authentication schemes. As discussed above, the current application of RFID tags in identifying failed components with long component information may be exposed to the risks of cloning and counterfeiting security attacks. For high-value components, developing a higher-secured RFID system that includes an encrypted algorithm and dual authorization is justified. These proposed enhancements will expand the application of reverse logistics with IoT and BI for sustainable service parts management.

**Author Contributions:** Conceptualization, D.Y.M. and C.Y.T.M.; methodology, D.Y.M. and C.Y.T.M.; software, C.Y.T.M.; validation, D.Y.M., D.C.K.H. and Y.W.; formal analysis, D.Y.M. and D.C.K.H.; investigation, D.Y.M. and Y.W.; writing—original draft preparation, D.Y.M., C.Y.T.M. and Y.W.; writing—review and editing, D.Y.M. and D.C.K.H.; funding acquisition, D.Y.M. All authors have read and agreed to the published version of the manuscript.

**Funding:** The work described in this paper was partially supported by a grant from the Research Grants Council of the Hong Kong Special Administrative Region, China (Project No. UGC/FDS14/E05/21).

**Institutional Review Board Statement:** Not applicable.

**Informed Consent Statement:** Not applicable.

**Data Availability Statement:** Not applicable.

**Acknowledgments:** We would like to thank three anonymous reviewers for their constructive comments during the review process and the support from the Big Data Intelligence Centre (BDIC) at the Hang Seng University of Hong Kong.

**Conflicts of Interest:** The authors declare no conflict of interest.

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
