# Peer review of "Design of a Reverse Logistics System with Internet of Things for Service Parts Management"

_sustainability, doi:10.3390/su141912013_

Round 1
Reviewer 1 Report
The manuscript, titled as "Design of Reverse Logistics Systems with Internet of Things for Service Parts Management", did presentt an interesting and valuable research study by leveraging the state-of-the-art technologies, including IoT and BI. Some merits can be found in this research study, but some improvements should be made to enhance the quality and research rigour. Authors may refer to my comments as follows:
- Manuscript title is a bit weird, namely "Design of Reverse Logistics Systems..." Does it mean a number of systems to be designed in this study?
- It would be better if this manuscript can be proofread or edited to address some grammatical mistakes.
- Managing serviceables and unserviceables is a common industrial practice. The contributions of this study can be emphasized in Section 1.
- In Section 2.2, the reviewed IoT/BI technologies are relatively outdated. Authors may consider to refer to some recent IoT/BI development in the logistics and supply chain management, other than OLAP and RFID , for example:
Integrating Internet of Things and multi-temperature delivery planning for perishable food E-commerce logistics: a model and application. International Journal of Production Research, 59(5), 1534-1556. (2021).
The business model of intelligent manufacturing with Internet of Things and machine learning. Enterprise Information Systems, 16(2), 307-325. (2022).
A Blockchain-IoT Platform for the Smart Pallet Pooling Management. Sensors, 21(18), 6310. (2021).
The contribution of Industry 4.0 technologies to facility management. International Journal of Engineering Business Management, 13, 18479790211024131. (2021).
- In Section 3.1, the overall information flow is clear to show the essential components, namely RFID and cloud, but it would be much better to explain perception, network, service, and application layers under the service-oriented architecture.
- Discussions after the case study in Section 4 should be provided, at least managerial implications. It is optional for authors to suggest and proposed the intended benefits from the IoT/BI implementation in the reverse logistics process.
Author Response
Thanks for your constructive comments to further enhance the quality of manuscript. Please see the attachment of our responses.

Reviewer 2 Report
No changes are required. the article is acceptable in its present form
Author Response
General comments: No changes are required. The article is acceptable in its present form.
Response: Thanks for the reviewer’s time and the positive assessment.

Reviewer 3 Report
The paper addresses a very interesting and appealing research topic; the approach developed is quite clear, and the research contents are very interesting. Nevertheless, in my opinion, the following issues have to be addressed before considering the article for publication:
• According to the authors, the strategies proposed by the model led to “sustainable” operations. This goal is inconsistent with the results achieved, in which the evaluations have been conducted exclusively from an economic perspective. In my opinion, defining the approach as “sustainable” is not possible. Consistent with this comment, please include an evaluation of the environmental and social gains ensured by the research work developed.
• Let me suggest stating the paper's aim in the first part of the manuscript (for instance, in the “Introduction” section).
• The authors stated that a decision model was developed to identify the failed components. In my view, the paper's approach is inconsistent with this goal; in the research work developed, the model does not seem to support any decision-maker. For instance, no interaction was identified between the decision model and the user, the input/output parameters considered by the decision model were neglected, the decision criteria were not detailed, etc. Therefore, let me suggest modifying the paper framework consistent with this view, or please describe the model as an “analytical model” (or procedure).
• Please check on the author's guidelines and the reference style of the text. I fear that it is not possible adopting bold font in the body of the manuscript (for instance, as shown on page 7).
• The meaning of function cost (i.e., “Refurbishing cost”, “Disposal cost”, and “Profit”) showed in EQs. 1-3 should be included in the body of the manuscript rather than in the equation line. Moreover, please avoid showing three dots before the equation identification number.
• In equations 4-7, I suggest removing the intermediary analytical steps to identify the final relations.
• In line 311, it is not necessary to show equation 4 in the body of the manuscript.
• There are no units of measure of the values introduced in table 1.
• The label shown on the x-axis of figure 6 is unclear; what means A=B? Please check or detail this label.
• Please adopt the black font in the graphs shown in figs 6-8 and remove the boundary line. Moreover, on the y-axes (consistent with the values on x-axes) should be shown the “unit” profit and cost.
• Please detail the final part of the manuscript (conclusion section), highlighting the limit and the future development of the research work conducted.
Author Response
Thanks for your constructive comments. All the issues have been addressed. Please see the attachment.

Round 2
Reviewer 3 Report
The paper addresses a very interesting and appealing research topic; the approach and the topics discussed in the paper are new and justify the interest for the publication. The structure of the paper is correct. The revisions adopted have improved the work. The suggestions proposed in the review report are included in the last draft of the paper. The major gaps filled by the manuscript are now described in “Introduction” and “Conclusions”. A new "Discussion" section has been introduced. The mistakes identified in the previous draft of the paper have been corrected.
Good luck!